# Charcoal as an Energy Resource: Global Trade, Production and Socioeconomic Practices Observed in Uganda

Catherine Nabukalu * and Reto Gieré *

Department of Earth and Environmental Science, University of Pennsylvania, 240 S. 33rd Street, Philadelphia, PA 19104-6316, USA
* Correspondence: catn@sas.upenn.edu or catn0371@gmail.com (C.N.); giere@sas.upenn.edu (R.G.);
  Tel.: +1-267-356-0371 (C.N.); +1-215-898-6907 (R.G.)

**Abstract:** Around the world, charcoal has persisted as an energy resource and retained unequivocal dominance in the energy consumption mix of some nations many years on since modern alternatives were invented. Furthermore, it has secured unyielding significance as a commodity on local and international markets and remained an aggressive competitor to electricity and gas for cooking. Here, we analyze the charcoal supply chain and highlight the rudimentary production techniques common within the sub-Saharan region, using Uganda as an example. Top global producers, importers, and exporters are discussed and, based on fieldwork from ten locations in Uganda, we describe common trade practices, economic contributions and the realities of charcoal consumption in areas with concentrated grid and electricity coverage. Indeed, forest degradation and deforestation in the charcoal trade is indiscriminate and the world's top producers and exporters of charcoal do not necessarily have vast forest resources. Pyrolysis, the process used to produce charcoal from wood, exacerbates risks of wild fires and deteriorates air quality. Our fieldwork indicates that little to no innovation exists to manage waste materials such as ash and polluting gases along the supply chain. Recommendations for the future include better forest conservation practices and more innovation at the cooking level, because effects of localized environmental degradation inevitably lead to negative impacts beyond geographical borders.

**Keywords:** charcoal production; import/export; cooking; deforestation; earth-mound kilns; electricity; environmental degradation; nomadism; public health

## 1. Introduction

All over the world, demand for energy has exacerbated human dependence on natural resources [1]. According to the United Nations' Food and Agricultural Organization (FAO), more than 50% of wood from forests worldwide is used for energy production [2]. In 2017, 51.2 Million tons (Mt) of wood charcoal were produced globally, up from 37.0 Mt in 2000 [3]. From 1993 to 2017, the largest average amounts of charcoal were produced annually in Africa (24.6 Mt), with 57% of the global production, followed by the Americas (23%, mostly Latin America), and Asia (18%; Figure 1a). Charcoal production is associated with environmental degradation, including deforestation [4], which is highest in Africa, followed by Latin America and the Caribbean [5].

In Europe and North America, consumption of charcoal is less conspicuous, but present nonetheless. For example, charcoal is used extensively as leisure fuel (e.g., for barbeques) in some industrialized countries [6,7]. Limited local production (Figure 1a) is made up for with imports of charcoal to meet existing demand. Indeed, 40% of the charcoal used in Europe is imported from Africa, with

Nigeria, Egypt, Namibia, and South Africa as key players [6,8,9]. Intra-European charcoal trade also exists, with Ukraine, Lithuania, and Latvia as main suppliers to Belgium, Germany, and Poland [6,9]. The use of charcoal as a source of energy or electricity is typically concealed in some regions because it is overshadowed by other dominant sources such as coal and gas in the energy mix [10]. Charcoal consumption is often linked to lack of modern alternatives but all over the world, electrified regions are still using charcoal as an energy source. In Brazil, the world's largest producer (Figure 1b) and consumer of charcoal, for example, more than 90% of the population has access to electricity, yet residential consumption has persisted at 9.7% of the country's total charcoal production [11]. Mainland China also features in the world's top 10 charcoal producers (Figure 1b) despite achieving widespread access to electricity [3,12]. Moreover, Germany, which has a diverse portfolio of modern energy resources [13], is still the world's biggest importer of charcoal [3] (Figure 1d).

Likewise, in Africa, the ubiquity of charcoal [3] (Figure 1a) and its unequivocal dominance in the energy-consumption mix despite rising incomes [14,15] conceal the increasing availability of electricity in various cities or countries and the reliance on other energy sources, such as direct sunlight. This fact partly explains why large parts of Africa are reported to be "in the dark" or "without energy" [16,17], even where electricity and charcoal are used interchangeably based on preferences. In Egypt, for instance, more than 88% of the population has access to electricity, but the country remains among the world's top charcoal producers today (Figure 1b) and an active importer within the past decade [3,12]. Similarly, Uganda's capital, Kampala, has considerable access to electricity, with more concentrated grid connections relative to other parts of the country [18,19], yet we observed during our field work that it is also one of the largest markets for the country's charcoal produced on the islands in Lake Victoria and in rural areas [20]. Whereas charcoal production occurs throughout the country, it is more concentrated near populous urban centers with lucrative markets, such as Kampala, Jinja, and Entebbe (Figure 2), because demand is mainly driven by urban centers [21,22]. This trend has also been observed in other countries [23–26]. Charcoal consumption in Uganda is still extensive alongside other biomass (e.g., firewood), taking a 92% share in the local energy-consumption mix [4,20,27], even though the country has scaled up investments in rural electrification [28].

The charcoal trade is highly informal [15,21,29], characterized by low record keeping and little to no tracking of the source of inputs [6] and waste materials along the value chain. Global charcoal imports and exports are estimated at US$1.16 B [30]. From 1993–2017, the world's top 10 charcoal-producing countries (Figure 1b) generated an average of 24.5 Mt of charcoal annually, of which more than 50% were produced by Brazil, Nigeria, and Ethiopia [3]. With the exception of Nigeria and Mainland China, top producers are not necessarily the leading exporters of charcoal. The US$784 M charcoal exports [31] are mainly sourced from the tropical rainforests of Indonesia (Figure 1c). Despite having relatively scant forest areas, Somalia, Argentina, Cuba, Poland, and Namibia are also important exporters. From 1993–2017, the top 10 exporters altogether shipped on average over 880 kilotons (kt) of charcoal per year [3]. Incidentally, some of the countries with low risk to energy security, defined as the continuous availability of energy at an affordable price, including Germany, Japan, France, and the UK [13,32], are among the top importers of charcoal (Figure 1d). For instance, of the 820 kt of charcoal received by the top 10 importers (annual average 1993–2017), Germany (153 kt) is by far the largest importer [3].

In the same way as highly electrified countries still heavily import charcoal and are considered to have abundant energy supply [6,13] (Figure 1d), reliance on charcoal as a fuel and its ubiquity in sub-Saharan Africa [33] should not by itself signal lack of alternatives. Moreover, in areas where electricity is indeed scarce, such a reality does not amount to absence of light or energy. The main objective of this study is to evaluate charcoal as both an energy source and a commodity. The paper describes some methods of extraction and transportation within the sub-Saharan region. The status and the future of the charcoal trade in the global supply chain are discussed. With Africa as the main producer of charcoal, Uganda is used as a case study to detail rudimentary production, trade, and local

market practices, and to give insight into demand for charcoal compared against some existing energy alternatives (e.g., electricity).

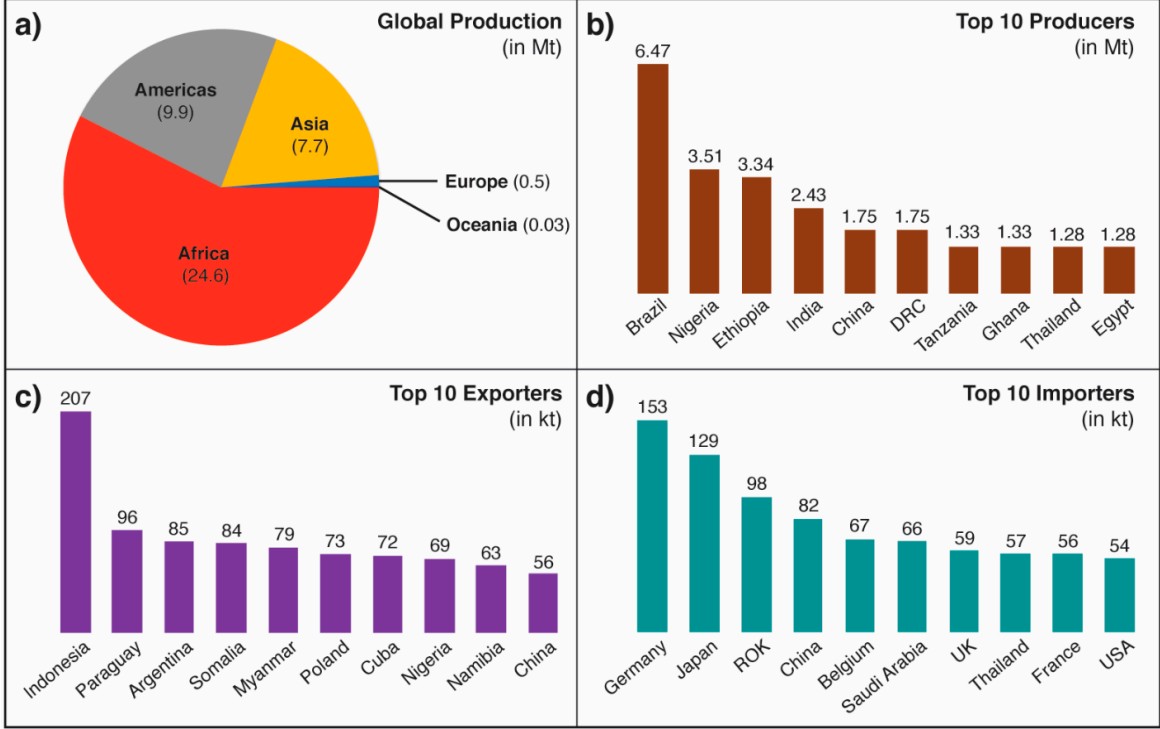

**Figure 1.** Global charcoal production and trade. (**a**) Production by region; (**b**) top producers; (**c**) top exporters; (**d**) top importers. DRC = Democratic Republic of the Congo; China = Mainland China; ROK = Republic of Korea. All data represent annual averages 1993–2017 [3].

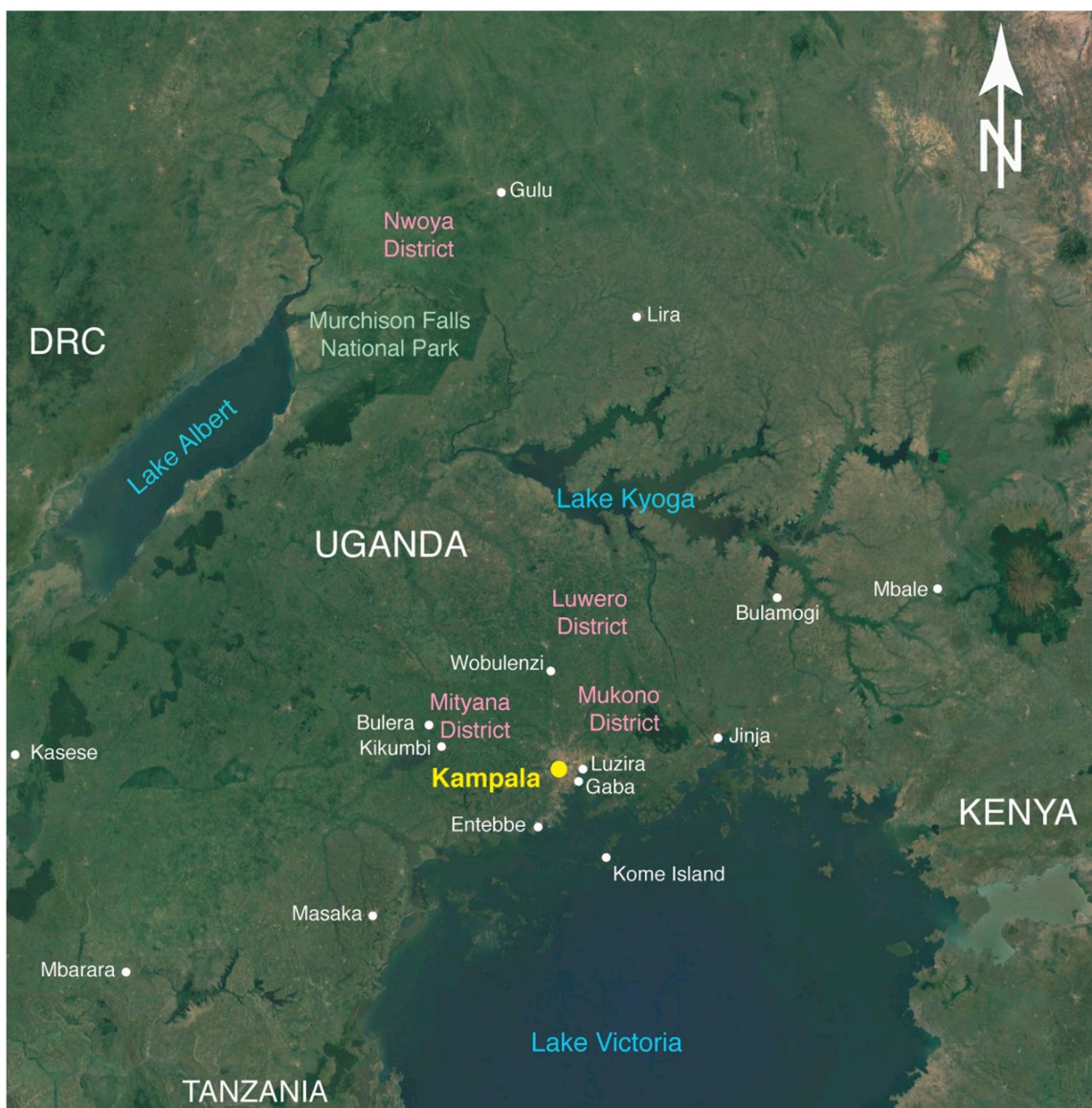

**Figure 2.** Satellite image showing the location of Uganda's capital Kampala and of other major towns as well as the districts and/or some of the sites investigated during this study. Image from Google Earth 2018 (Image Landsat/Copernicus). DRC = Democratic Republic of the Congo.

## 2. Materials and Methods

### 2.1. Data Collection Method

All data were collected in October 2017 during field work in Uganda. Key stakeholders were interviewed at their workplaces, and included the following: 18 charcoal producers, 3 loaders/off-loaders, 3 final traders, and 4 logistics personnel (Table 1). All interviews were informal but designed for learning about, and documenting, today's charcoal production and trade in Uganda. Interviewees discussed their work and common practices in charcoal production and trade. Some interviewees further shared details about their family life and their perception of the status and future of this trade, with respect to regulation, environmental degradation, and the critical role of charcoal in energy markets. Driving forces and limitations for their involvement in the trade were discussed and, while most participants spoke about their own experiences, some had been in the trade for so long, over 20 years, that they could provide a valuable general picture of what is typical in the supply chain. Moreover, their knowledge of other stakeholders allowed for discussion

on experiences beyond their own. Photographs presented are our own, and people featured were actual charcoal retailers or producers. The information collected during the interviews, as well as our own observations and research in the field, provided the basis for Section 3, Results and Discussion, where we describe the results and the significance of our data and observations and compare them with available literature data.

All mentioned tree species are identified by their scientific names (in Italics), with their equivalent Luganda (in quotation marks) and English names in parenthesis.

## 2.2. Study Areas

A total of 10 sites in the Central and Northern regions of Uganda were visited during the field work (Table 1). Eight of these sites were located in rural areas (Figure 2): (a) Bulera-Busaana, Kikumbi, and Namiwunda in Mityana District; (b) Gaba, a community on the shore of Lake Victoria, where charcoal is unloaded from boats arriving from islands (e.g., Kome) in the Mukono District; (c) Kigoloba, Nsero, and Wobulenzi in the Central Luwero District; and (d) Naminato Bridge in the Northern Nwoya District. These rural sites were selected to allow for a comparison between charcoal production and trade practices in the Northern and Central regions of the country (see Table 1), which exhibit substantial differences in vegetation (Table 1): briefly, the Central region is characterized by semi-deciduous forests and woodland savanna, with mostly soft-stem trees, whereas the Northern region comprises dry, wooded, and grassland savanna. The differences in vegetation among the studied field sites influence the tree species available for local production and the perceived charcoal quality. Nonetheless, cross-district trade makes charcoal from many tree varieties available in populous cities, including the capital of Kampala.

Two of the studied localities were urban sites, which were located along McKay Road and Wakaliga Road in Kampala (Table 1). These sites were chosen at random in order to study the charcoal trade in Kampala, because charcoal is sold along many roads in the city.

Table 1. Summary of results obtained at the study sites in the Central and Northern regions of Uganda.

| District and Study Site | Interviewees and Their Major Role in Supply Chain | General Characteristics of Vegetation Described in [34] and Samples Collected during Our Fieldwork | Discussion Unique to Site (Production, Trade, and Socioeconomics) | Comments and Alternative Use of Trees (from Our Own Observations and from [34]) |
|---|---|---|---|---|
| **Central Region** | | | | |
| **Kampala District** — Gaba, shore of Lake Victoria | 3 loaders/off-loaders 1 logistics personnel (between Kome–shore) 2 final traders loading/off-loading from boats final trading logistics | Semi-deciduous forest on often cultivated land, woodland savanna, e.g.,: <ul><li>*Bridelia micrantha* ("Katazamiti", Bridelia or coast goldleaf)</li><li>*Coffea arabica* ("Kaawa", Coffee)</li><li>*Eucalyptus citriodora, Eucalyptus grandis, Eucalyptus camaldulensis* ("Kalitunsi", Eucalyptus)</li><li>*Ficus natalensis* ("Mutuba", Natal Fig)</li><li>*Harungana madagascariensis* ("Mulilira", Haronga)</li><li>*Markhamia lutea* ("Musambya", Nile Tulip)</li><li>*Musanga cecropioides* ("Kaliba", Umbrella tree)</li></ul> | <ul><li>Safety of life at sea, boat traffic, schedules, and encounters with lake police while transporting charcoal, connecting with island producers</li><li>Being the middleman between producers on the island and customers at the shore</li><li>Trader's perception of lake traffic police, regulation and enforcement for charcoal bans, legal trade of charcoal, and licensing for spots at the market place</li><li>Packaging of the final products and display as a form of advertising at market stalls</li><li>Motivations and barriers for engaging in the charcoal trade</li><li>Traders' perception of electricity as a competitor to charcoal</li><li>Quality management for traders that are not involved (or invested) in production</li><li>The role of mobile phones and online/virtual banking systems as facilitation in charcoal trade</li><li>Home deliveries</li></ul> | Samples include vegetation sourced from islands of Lake Victoria, such as Kome, which has softer stems. Charcoal from island species is mixed with that from tree species at the shore (mostly from within the Central region). tree species are medicinal. For example, the *Vernonia amygdalina* ("Mululuza", Bitter Leaf) is a remedy for malaria in the Central region. Other uses: Firewood, timber, poles (granaries, power lines), tool handles, fruit, medicine (bark, leaves, roots), fodder (leaves), mulch, shade; beverage, posts, timber (construction), bee forage, ornamental, windbreak, live fence, bark cloth, ornamental, shade timber (boxes, crates), carving (utensils, musical instruments), beehives, mulch. |

**Table 1.** *Cont.*

| District and Study Site | Interviewees and Their Major Role in Supply Chain | General Characteristics of Vegetation Described in [34] and Samples Collected during Our Fieldwork | Discussion Unique to Site (Production, Trade, and Socioeconomics) | Comments and Alternative Use of Trees (from Our Own Observations and from [34]) |
|---|---|---|---|---|
| McKay Road, Nateete | 1 roadside seller (also, business owner) | Semi-deciduous forest on often cultivated land, woodland savanna, e.g.,:<br>• *Combretum apiculatum* ("Nkoola", Red bushwillow)<br>• *Combretum spp* ("Ndagi", Bushwillow)<br>• *Myrianthus holstii* ("Kigungu", Soup tree)<br>• *Zanthoxylum gilletti, Zanthoxylum rubescens* ("Munyenye", Prickly ash or Hercules club) | • Trading practices, preference in original species for charcoal<br>• Species and the impression of charcoal quality<br>• Measuring instruments of charcoal for pricing to the final consumer<br>• Negotiation with first-tier suppliers<br>• Quality management and product handling, including packaging and storage<br>• Waste disposal for charcoal silts<br>• Licensing for trading in decentralized markets and dealing with market operators<br>• Unloading charcoal and upstream procurement<br>• Determining 'value for money' to buyers | Tree species found here were sourced from many different regions.<br>Final market: no production was observed here.<br>Traders bought this charcoal from large truck off-loaders already in the city.<br>Other uses: Firewood, poles, posts, timber (construction), tool handles, bee forage, mulch |
| Wakaliga Road, Nateete | 1 producer and trader (owns means of production and transportation, oversees 4 staff at final market stall) | Semi-deciduous forest on often cultivated land, woodland, savanna, e.g.,:<br>• *Combretum apiculatum* ("Nkoola", Red bushwillow)<br>• *Combretum spp* ("Ndagi", Bushwillow) | • Coordinating logistics, trade, transportation,<br>• Supply chain management<br>• Negotiation with multiple stakeholders<br>• Unloading trucks in the final marketplace<br>• Trade in the final market and valuation of market-ready commodities<br>• Charcoal as a structured trade, including recruitment and workforce management<br>• Measuring instruments in the final market | There was preference to sourcing 'Northern species', i.e., *Combretum spp* ("Ndagi", Bushwillow) because they can be sold for higher prices for perceived better quality.<br>Tree species were sourced from many regions.<br>Other uses: Firewood, timber (furniture, construction), tool handles, flooding), medicine (bar) |

**Table 1.** *Cont.*

| District and Study Site | Interviewees and Their Major Role in Supply Chain | General Characteristics of Vegetation Described in [34] and Samples Collected during Our Fieldwork | Discussion Unique to Site (Production, Trade, and Socioeconomics) | Comments and Alternative Use of Trees (from Our Own Observations and from [34]) |
|---|---|---|---|---|
| **Luwero District** Kigoloba | 1 charcoal burner (also land owner of site), production and direct delivery to private residences | Soft-stem trees of the central region. Also found on the islands near Lake Victoria, e.g.,: <ul><li>Albizia coriaria ("Mugavu", Welw. ex Oliv)</li><li>Grewia mollis Celtis mildbraedii ("Nkomakoma/Mukomakoma", Natal white stinkwood or Natal elm)</li></ul> | <ul><li>Charcoal production and opportunities around burning charcoal as a secondary activity for income</li><li>Sale of charcoal to first-tier buyers from the kiln and delivery to roadsides to attract new markets</li><li>Tree species suitable for charcoal</li></ul> | Samples also found in central region islands. Trees were sourced locally from Kigoloba. The producer is a small-scale farmer that cut trees from his private land. Other uses: Firewood |
| Nsero | 1 charcoal burner (also land owner of site) production and direct delivery to private residences | Soft-stem trees of the central region, e.g.,: <ul><li>*Artocarpus heterophyllus* ("Fene/Mufenensi", Jackfruit)</li><li>*Markhamia lutea* ("Musambya", Nile tulip)</li><li>*Ficus natalensis* ("Mutuba", Natal Fig)</li></ul> | <ul><li>Charcoal production on private land as a source of secondary income to the farmer, trade, and e-payments</li><li>Regulatory trends over the past 20 years by the village council</li><li>Accessibility of production sites using footpaths</li><li>Set up of the "Kasisira" method and its contrast to the "Baasi" and "Kajegere" methods</li><li>Charcoal production near burners' residences</li></ul> | Trees were sourced locally from Nsero Producer is a farmer that cuts the fruit trees from his private land. *Artocarpus heterophyllus* ("Fene/Mufenensi", Jackfruit) has high sap content, and therefore higher chances of poor pyrolysis Other uses: Firewood, timber (furniture, carts, lorry bodies, doors), food (fruit, seed). |
| Wobulenzi (along Gulu–Kampala highway) | 3 truck drivers (team) i.e., logistics personnel | Not applicable because research at this site was on Kampala–Gulu highway, about transportation and logistics practices | <ul><li>Truck loading and overloading</li><li>Scheduling for driving and delivery to key markets</li><li>Encounters with road safety and traffic police</li><li>Overnight transportation of charcoal</li></ul> | Overloading is a norm, and regulators have insufficient facilities to enforce safety. |

**Table 1.** *Cont.*

| District and Study Site | | Interviewees and Their Major Role in Supply Chain | General Characteristics of Vegetation Described in [34] and Samples Collected during Our Fieldwork | Discussion Unique to Site (Production, Trade, and Socioeconomics) | Comments and Alternative Use of Trees (from Our Own Observations and from [34]) |
|---|---|---|---|---|---|
| Mityana District | Bulera-Busaana | 3 producers (local nomads, same team and family) | Semi-deciduous forest on often cultivated land, woodland savanna, e.g.,: <br>• *Amaranthus* ("Nakati", Spinach) (grown by nomadic charcoal burners) | • Nomadic production and its challenges <br>• Land use after harvesting charcoal <br>• Economic opportunities, production in rural areas <br>• Product packaging and storage at the kiln <br>• Contacting first-tier buyers to initiate wholesale trade of charcoal <br>• Ownership of charcoal (landlord vs. charcoal burners) <br>• Accessibility of land in nomadic charcoal production | Nomads belonged to the same family but were from different generations. <br>Female participation in charcoal production <br>The *Amaranthus* was grown here by the nomadic charcoal burners on spots that were previously occupied by earth-mound kilns (same observation with Amaranthus in Nwoya District, near Naminato bridge). |
| | Kikumbi | 5 charcoal producers (team) and transporters (short-distance nomads) | Semi-deciduous forest on often cultivated land, woodland savanna, e.g.,: <br>• *Mangifera indica* ("Muyembe", Mango) <br>• *Psidium guajava* ("Mupeera", Guava tree) | • Nomadic production and safety techniques during production <br>• Age of the vegetation used in production processes <br>• Transportation permits from local authorities for charcoal-only transit <br>• Tree species suitable for charcoal | All local charcoal-only transporters require permits. <br>Nomadic charcoal burners, trees cut have relatively soft stems, average age of trees was around 30 years. |
| | Namiwunda | 1 producer (not landowner of site), production and direct delivery to private residence | Semi-deciduous forest on often cultivated land, Woodland savanna, e.g.,: *Musanga cecropioides* ("Kaliba", Umbrella tree) | • Very remote and hilly site <br>• Challenges for transportation of charcoal using motorbikes <br>• Inaccessibility due to poor roads <br>• Difficulty of oversight by regulators <br>• Tree species suitable for charcoal | Negotiation between landowners and charcoal burners to conditions of permitting production. <br>Charcoal was produced with local trees. <br>**Other uses:** Shade, firewood, fruit trees |

**Table 1.** *Cont.*

| District and Study Site | Interviewees and Their Major Role in Supply Chain | General Characteristics of Vegetation Described in [34] and Samples Collected during Our Fieldwork | Discussion Unique to Site (Production, Trade, and Socioeconomics) | Comments and Alternative Use of Trees (from Our Own Observations and from [34]) |
|---|---|---|---|---|
| | | **Northern Region** | | |
| **Nwoya District** Naminato Bridge | 6 charcoal burners (team), mass producers (long-distance nomads) | Dry savanna vegetation, wooded savanna, and savanna grassland, e.g.,: <br>• Albizia glaberrima, Albizia zygia, Albizia grandibrac Zteata ("Nongo", Silk tree) <br>• Amaranthus ("Nakati", Spinach) <br>• Ficus natalensis ("Mutuba", Natal Fig) <br>• "Tooke Kulu" | • Tools used for deforestation <br>• Overview of upstream charcoal bans, burning on private farmlands; market practices at the kiln, including negotiation and wholesale charcoal trade with first-tier buyers, access of large motor vehicles to semi-forested regions <br>• Labor relations, including language barrier at work for nomads in 'foreign' ethnic regions <br>• Informal recruitment of charcoal burners for skills <br>• Sustenance at work sites: temporary housing and healthcare for the duration of work, regeneration of vegetation at remote sites | *Amaranthus* ("Nakati", Spinach), a dietary supplement, was grown here by the nomadic charcoal burners on spots that were previously occupied by earth-mound kilns. <br>Other uses: Firewood, edible wild fruits |

## 3. Results and Discussion

### 3.1. Deforestation for Charcoal Production

Because most trees can be used to make charcoal [34], deforestation is indiscriminate in this trade. Even though Sedano et al. [25] and Namaalwa et al. [4] discuss the notion of harvesting by "minimum harvestable diameter" [4], we found at the studied sites that determining tree diameters was not practiced prior to felling trees. Indeed, Van Beukering et al. [23] observed that, due to poor management, young trees were cut to cover earth-mound kilns. Moreover, the thin branches of young trees are - easier to set alight (see Figure 3d). The speed of cutting trees depends on tools used by the burners rather than forest density. Recognizing that application of machinery accelerates deforestation compared to manual cutting, our research showed that the use of chain saws has been made illegal in many parts of Uganda, including the Nwoya and Mityana Districts, although enforcement is hard [15].

At our study sites, we observed that most trees were felled manually, using hand tools such as axes or pangas. A single axe with a sharpened edge is swung to cut a tree from the bottom of its trunk. Once the tree hits the ground, pangas are used to cut the branches to manageable sizes that can easily be arranged to form the earth-mound kilns. The tree stumps are usually abandoned at the site to regenerate and sprout into newer vegetation, which may lead to preservation of vegetation in tropical areas and woodlands [4,11]. Nevertheless, we observed some use of machinery in the Northern Nwoya District and in Namiwunda township (Mityana District). Machinery operated in deforestation is often powered and lubricated with flammable fuels (e.g., oil, kerosene), and thus operating them in forested areas, where fires get set in earth-mound kilns for charcoal production, creates risks of wild fires.

Our research revealed that in the downstream supply chain, it is difficult to identify the origin of trees that were cut to produce charcoal. In a national market place like Uganda, for example, traders trace the origin of charcoal by visual inspection and knowing from experience the characteristics of charcoal derived from typical vegetation in specific regions. Even then, one would not easily state the precise location of these forests if one were not directly part of procurement in the upstream stages, because forests are typically private and rarely mapped with sophisticated zoning tools. Similarly, and according to the WWF [6], charcoal in Germany is mainly from non-certified tropical forests. Moreover, the packaging that states its origin still lacks precision, and suppliers are unwilling to be transparent about procurement practices. This makes tracking deforestation activities difficult and hinders consumer education about the origin of this 'barbecue' fuel [25]. Indeed, most charcoal supplied by Dancoal, Germany's largest supplier, is sourced from countries where actions against deforestation are limited, even where legislation is present to render it illegal for charcoal production, such as Nigeria [6].

Deforestation is occurring faster than some countries have natural forests to accommodate it. For instance, of Egypt's total land area (1.01 Million km$^2$), only 0.1% is covered by forest, and yet Egypt is one of the top 10 charcoal producers (Figure 1b). Brazil, DRC, and Tanzania have forest-to-land area ratios close to, or greater than 1, but several of the top charcoal producers have considerably less forests [5]. Altogether, the top producing countries have over 2700 threatened tree species [3,5], and whereas the loss of these species is not entirely attributed to charcoal production, it is worth noting that practices in this trade lead to indiscriminate forest loss or, at the very least, forest degradation. The degradation and loss of forest areas are not only due to excessive charcoal production but also because the methods used to obtain charcoal create risks of wild fires that may wipe out vegetation extensively.

As natural forests speedily recede in some countries while others add charcoal to their already abundant energy mixes through imports [10], evaluation and implementation of better conservation practices in this trade are imperative. The available data demonstrate that charcoal is not only consumed in countries with less modern alternatives. Indeed, its demand and consumption sustain where access to electric grids and modern alternatives are present [18]. Moreover, localized environmental degradation in producer countries inevitably leads to negative impacts that extend outside geographical borders [35].

### 3.2. Natural Resources for Food Versus Fuel

According to our field observations, tree species, such as the *Psidium guajava* ("Mupeera", Guava tree), *Artocarpus heterophyllus* ("Fene", Jackfruit tree), and *Mangifera indica* ("Muyembe", Mango tree), are sources of food, yet their trunks are commonly used for charcoal production at the discretion of burners and landowners [26,34]. Similarly, the leaves of *Vernonia amygdalina* ("Mululuza", Bitter Leaf) are used for human consumption and as an herbal medicine in the Central region of Uganda (e.g., against malaria), but their trunks are also used by charcoal burners.

However, we observed that deforestation in some parts of Uganda is done to subsequently use the land for agriculture, which means that felling fruit trees for charcoal does not necessarily lead to food being foregone in the 'food vs. fuel' analysis. For instance, increased commercial rice farming in the Mukono District [36] necessitated deforestation in townships, such as Kome, to eliminate tree shade for the cash crop to flourish. This in turn increased the supply of logs, which then were channeled into charcoal production. Similarly, our research has shown that, as more of Nwoya's private farmers pursue growing food crops, such as cassava, they cut other trees, including *Albizia glaberrima* ("Nongo", Lowveld Albizia), to deter wildlife (e.g., monkeys) that come from the neighboring Murchison Falls National Park reserve because they are considered pests on their farmlands. Indeed, charcoal production is often a secondary reason for deforestation, especially when land is cleared for farming [1,37]. In their study of Mozambique, however, Sedano et al. [25] suggested that deforestation for the purpose of charcoal production can exceed land clearing for agriculture. This means that large-scale forest degradation for charcoal production can lead to deficits in food supply [37].

### 3.3. How Charcoal is Made

Charcoal is made by pyrolysis—i.e., heating under low-oxygen conditions—of solid biomass (e.g., tree trunks, branches) to remove moisture and other volatile components (such as methane [26,37–39]. Because many tree species can be used to make charcoal, logging is indiscriminate. However, in the visited locations, our research revealed that there is no clear-cutting, but rather, trees are felled one at a time. Even though most species have varying uses, charcoal production can be the priority (see Table 1 and [34]). Below, we describe the specific practices of charcoal production as observed during our field work.

After the individual trees have been felled, their trunks are manually gathered into a heap. The heaps of stacked tree trunks are then covered with smaller branches and dry leaves, which are easier to set alight, and finally with a top layer of damp soil, which retains heat within the heap during pyrolysis and minimizes access of oxygen to the tree trunks. At the visited sites, we were able to distinguish three methods of assembling the logs to form the earth-mound kilns.

Method 1 is known as "baasi" in Luganda, which translates to 'bus'. The tree trunks are arranged horizontally in layers, and all logs face the same direction (Figure 3a,b). The shape of the pile thus resembles that of a bus. The (main) opening of the heap is covered by dry plant material and/or plastic bottles or pieces of tire, which are used to light the heap from one end.

Method 2 is known as "kajegere" in Luganda, which translates to 'chain'. The main difference to the baasi method is that the logs are set up to appear like a chain, or grid, whereby subsequent horizontal layers are stacked at an angle relative to the underlying layer (Figure 3c,d), rather than parallel and in the same orientation used in the Baasi method. The interviewed charcoal burners did not report any differences in efficiency between the two methods. Rather, our interviews revealed that the two methods reflect personal preferences.

Method 3 is known as "kasisira" in Luganda, which translates to 'hut'. It produces a heap that resembles a hut (Figure 3e). The logs are piled up standing at an angle and are then covered by a layer of damp soil to form a mound. As the mound is cone-shaped, wooden bars or sticks are placed vertically on the sides of the heap, linked by basic materials, such as ropes and parts of dry plants, to prevent downward erosion of the damp soil placed on top due to the steep gradient (Figure 3e).

In contrast to the baasi and kajegere methods, the fire is applied at the center of the heap, rather than on one end.

Common to all three methods is that the stacks of logs are covered by damp soil, which is dug from around the heaps with hand tools, such as hoes, and fetched with spades. Small exhaust holes are manually created on the heap covers by piercing through the damp soil layer, whereby the number of exhaust holes is based on the burners' judgement. These exhaust holes serve to release moisture (Figure 3f) and to regulate heat through air circulation. For their propensity to retain high heat, plastic materials and small pieces of tire are typically inserted into the heap and covered with smaller branches and dry plant materials to spread flames within the heap. Matchsticks are used to set the heap alight.

Because lower heaps are easier to manage than high heaps, the charcoal burners typically pile the wood to shoulder height. However, during burning, the damp earth on top of the heap loses moisture, which leads to destabilization of the continuous soil layer and to moving of dry soil material downward into the heap to occupy gaps between the stacked wood. In addition, because the produced charcoal has a smaller volume than the original logs, the heaps are destabilized, which increases risks of collapsing during the process.

This disruption of the continuous soil layer enlarges the exhaust holes and may create new holes, thereby allowing for more oxygen to enter the heap, which increases the rate of combustion and thus creates risks of burning logs to ashes rather than transforming them into charcoal. As this is undesirable and causes economic loss, the goal for burners is to limit oxygen supply into the heap. To make monitoring easier the burners tend to build the earth-mound kilns close to their homes or they set up tents as close as possible.

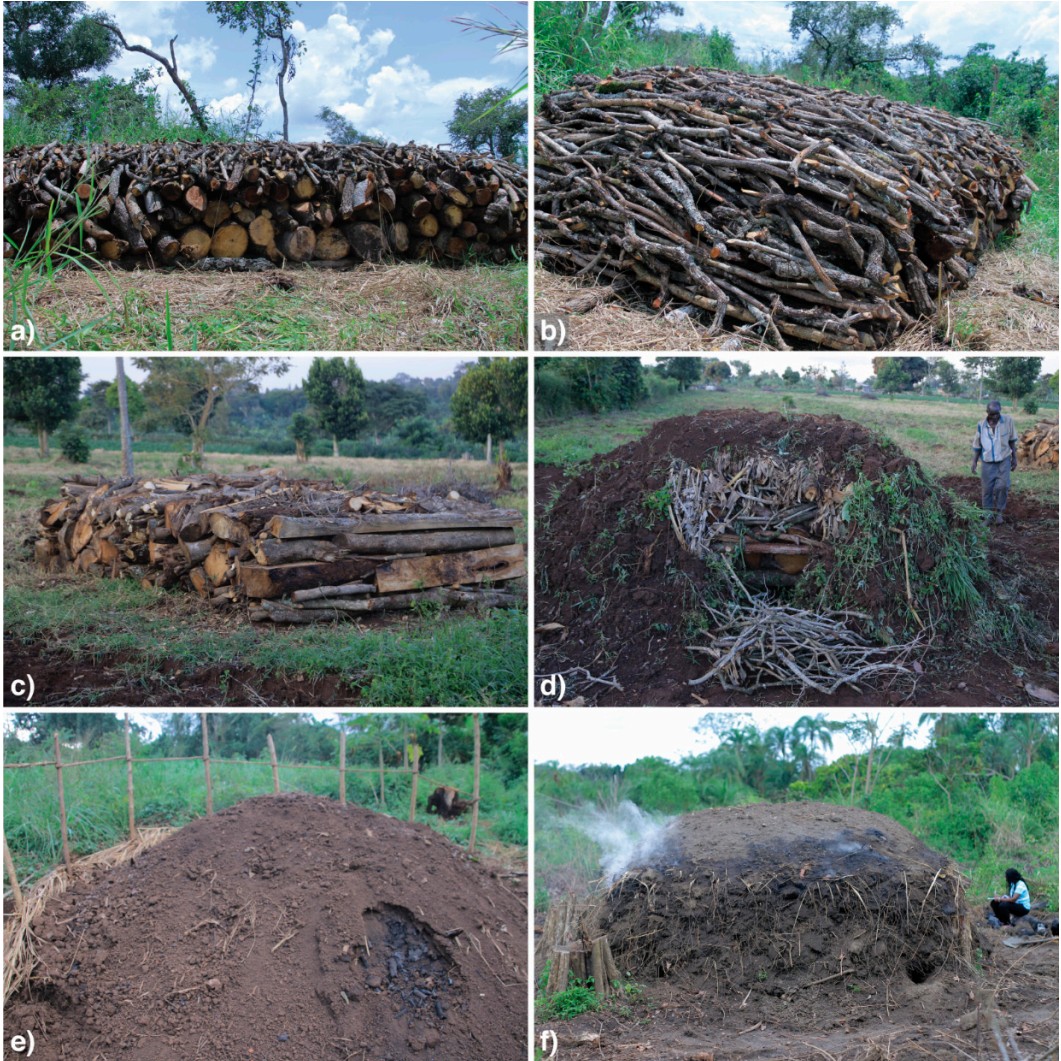

**Figure 3.** Earth-mound kilns in rural Uganda. (**a**) and (**b**) Baasi log layout, near Naminato bridge, Nwoya District; (**c**) Kajegere log layout, near Bulera, Mityana District; (**d**) Kajegere kiln with opening for setting it alight, near Bulera, Mityana District; (**e**) Kasisira kiln, near Nsero, Luwero District; (**f**) Kajegere kiln, showing steam and smoke escaping during operation, near Naminato bridge, Nwoya District.

*3.4. Nomadism in Charcoal Production*

Our field work revealed that most charcoal is produced in rural areas, even though firewood is used more commonly there. This result is in agreement with the work published by Tabuti et al. [20], who conducted their fieldwork in Bulamogi in the eastern part of Uganda (Figure 2). Indeed, Alfaro and Jones [26] also made the same observation in rural Liberia. A key result of our study is that nomadic charcoal production is an important characteristic of the charcoal supply chain in the investigated Central and Northern regions of Uganda. In practice, whereas some producers in Brazil [7,40] and Kenya [41] set up cemented kilns for more controlled processes, charcoal production as described above does not necessarily require sophisticated or stationary infrastructure. Our research showed that, when charcoal is the primary source of income, burners and entrepreneurs recognize the high economic expense of moving logs for long distances from the site of tree felling to a single production point. Tree stems are therefore transformed into charcoal very close to where the original tree grew (Figure 3f) to eliminate logistical costs of raw materials. Workers move closer to where trees are located as nomads and abandon areas with scarce vegetation. We observed that once production is finished, the burners move on to other areas, searching for new vegetation. However, this "spatial abandonment" [4] also

occurs in areas with dense vegetation, which are hard to reach and exploit, in many cases due to steep topography, which makes logistics wearisome.

In some cases, we learned from the charcoal burners that they were recruited for this purpose by private land owners or private informal companies. They were then transferred to various locations, with the precondition of staying there as long as the work lasts, typically 4–6 months. The nomads may travel in groups as employees or as independent entrepreneurs, and set up temporary waterproof tents for the duration of their work. Nomadism, to a large extent, is sustained because burning occurs mostly on private land where government oversight is limited. Indeed, over 47% of wood for Ugandan charcoal production is from private forests, whereas 16% comes from central government reserves [42]. The key challenge here is that being nomadic, charcoal burners do not own the land where they are contracted to work and, as a result, have no obligation for reforestation [43]. Moreover, in cases where reforestation programs are established, limited interest and funding were identified as key barriers [20].

Our interviews revealed that some landowners sell trees to charcoal burners and facilitate logging and charcoal burning on their premises. Because charcoal production requires an intimate knowledge of pyrolysis and combustion principles [23], entrepreneurial landowners hire burners for their skills to produce charcoal on their land, retaining sole ownership of the goods. Charcoal production in Uganda is banned in some districts [20] and transportation requires permits. However, social pressures can be a major barrier to limiting charcoal production [26]. Moreover, remoteness and nomadism make it difficult to track this activity, as also reported for the case of Mozambique [25]. Enforcement is mainly geared towards transportation practices although access to information is still equally limited in this sector. Moreover, political barriers, such as corruption and cartels, render central governments 'powerless' in the intervention to limit or ban charcoal production in the most upstream parts of the supply chain [15,23].

### 3.5. Harvesting, Waste, and Soil Quality at Earth-Mound Kilns

Our field work revealed that once the charcoal is formed and the earth-mound kiln has collapsed, it is ready for harvesting, which is done manually using hand tools. Indeed, some burners begin extracting charcoal on the collapsed end of the kiln while the other end keeps burning. During pyrolysis, logs lose moisture and the burners can tell when this moisture is exhausted, because of diminished smoke and vapor at the top of the kilns, a signal that most of the charcoal has formed. Because of the heat given off during the process, damp soil at the top of the kiln also loses its moisture and is transformed into loose silt, which slips downward into the cracks thereby cutting off oxygen supply. Burners do not open new exhausts on the heap to prevent combustion and ashing. Under oxygen-tight conditions, the fire suffocates and stops or travels within the heap to the end where logs are still burning, causing the charcoal on the collapsed end to cool off without the need to use water [20].

We observed that first, the top layer of soil is removed with spades to expose the charcoal underneath. Rakes are then used to collect the charcoal, separate it from silt, and to gather it for filling into sacks by hand (Figure 4a,b). Smaller pieces are sifted using sieves or picked up singly. In Uganda, white sacks made from woven polyethylene are commonly used in packaging because they are elastic, flexible, lightweight, inexpensive, and make handling easy. The sacks, usually about 120–180 cm tall, are filled with charcoal of varying weights and types to stand anywhere between 180–300 cm high, including artificial extensions to increase volume per sack (Figure 4b,c). Semi-charred logs are not uncommon at the production sites (Figure 4a). For instance, species such as the *Artocarpus heterophyllus* ("Fene/Mufenensi", Jackfruit), which naturally contain abundant sap in their stems, often come out semi-burnt, depending on, e.g., temperature, burners' tact, or experience [11]. The semi-burnt logs are either put through another pyrolysis cycle or abandoned at the site as waste. The remaining silts are often left bare at these sites. However, we observed that in Nwoya, near Naminato bridge, nomads used the spaces of former heaps to grow vegetables, such as *Amaranthus* ("Nakati", Spinach), and to supplement their diets during the periods spent in their remote worksites, because they consider them

to be highly fertile for agriculture due to presence of silted charcoal particles that enhance soil quality. This reflects the practice of using biochar as a soil amendment in many countries around the world [44].

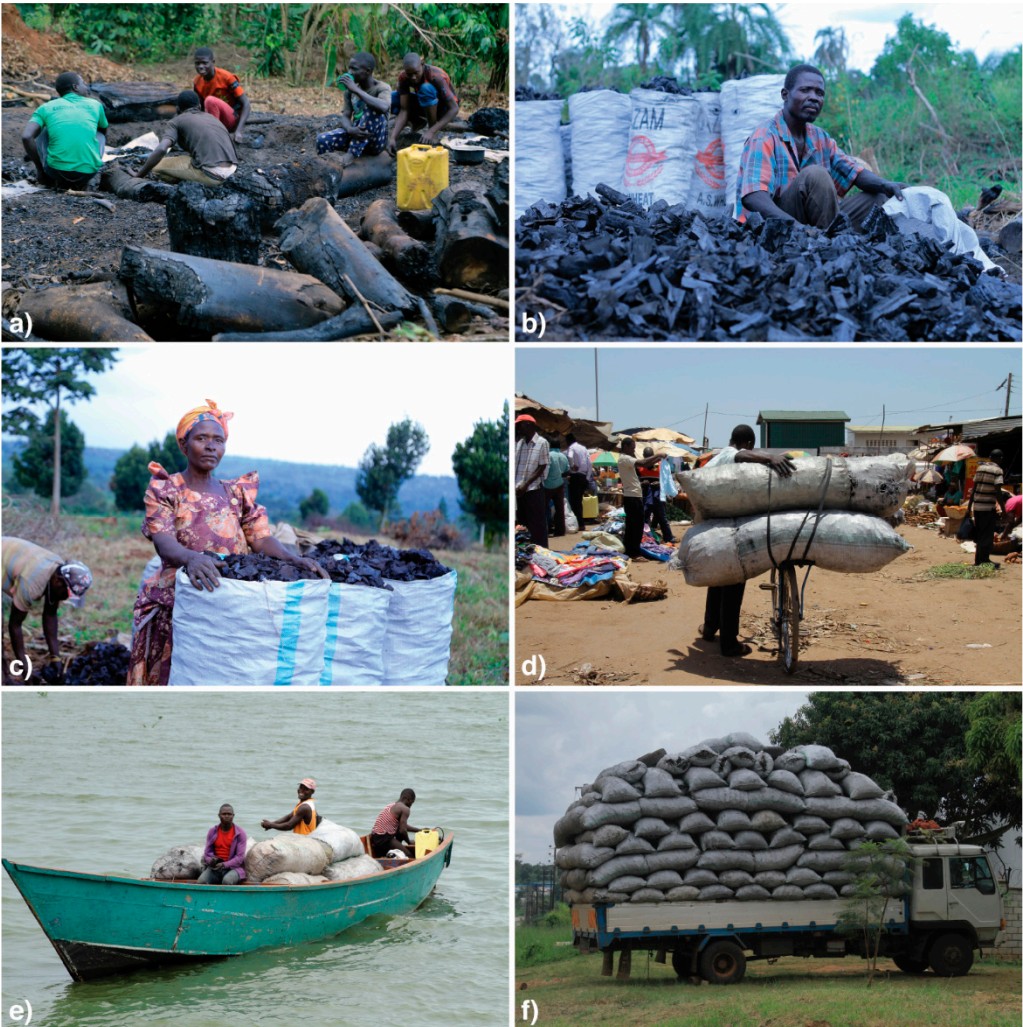

**Figure 4.** Harvesting, packaging, and transport of charcoal in Uganda. (**a**) near Kikumbi, Mityana District; (**b**) near Naminato bridge, Nwoya District; (**c**) near Bulera, Mityana District; (**d**) Gaba market, shore of Lake Victoria; (**e**) charcoal from Kome Island arriving in Gaba, shore of Lake Victoria; (**f**) charcoal truck waiting for overnight trip in Wobulenzi, Luwero District.

## 3.6. Transportation of Charcoal

In the areas visited during our field work, less developed feeder roads, typically made of murram (lateritic gravelly material), link remote communities to well-developed tarmac roads or highways. The murram roads are mostly narrow and in countless cases connected to even smaller footpaths. Many of these smaller transportation routes are not created by centralized government planning. Therefore, tracking charcoal transportation along such routes would be both wearisome and expensive for authorities. Moreover, as new paths are continuously created by charcoal burners to make work easier, transport links in this trade somewhat deviate from traditional ones, at least in the most upstream part of the supply chain. Goods are moved on bicycles, carts or pick-up trucks to bring them from more remote townships to roadsides, where they are purchased in small quantities by private customers [20,23] or wholesale by other traders.

In many cases, traders by the roadside are not producers [4]. However, we observed that some of these traders, being mainly local, have communication links to producers in remote areas in order to

maintain appropriate levels of inventory. Like specialists in any other skill or trade, some charcoal burners are known for their work by locals. They conduct relationship-driven business where customers reach them directly through cell phones to make deliveries straight to their premises. Light deliveries (1–3 sacks) are easily made using bicycles or motorcycles directly from kilns. Our investigations revealed that, from charcoal kilns to markets, between islands and shores to consumers' residences, charcoal may be transported by a range of parties in various ways, therefore intermodal or multimodal transportation is not uncommon in this trade. For instance, charcoal that leaves islands on engine boats is transported by bicycles, motorcycles, or human portage in final markets at the shores, depending on quantities purchased. Below is a summary of observations made during our field work (see also Figure 4).

Manual carts and bicycles are often used for short distances. Their role and intensity as a mode of charcoal transport should not be underestimated, however. They are easily affordable, and very flexible for navigating narrow ways such as footpaths and reaching remote production sites to carry charcoal. Moreover, if charcoal is moving from producers to final consumers, the chances of administrative oversight to monitor licensing are low. This is exacerbated by the fact that production occurs in remote rural areas with under-developed transport infrastructure, also observed by Shively et al. [43] in Uganda. These modes of transportation further play an important role in market places after trucks are unloaded. They carry goods to market stalls (Figure 4d) and to residences in case of bulk purchases, i.e., by the sack.

Motorcycles are much easier to use than manual carts and bicycles. They are faster, can carry heavier loads, and access steep remote areas. Even though they cost more, they are as common and accessible as bicycles. Ownership and operation of motorcycles is usually registered at the local, decentralized departments of transport. Many motorcycles are registered for transportation of both goods and passengers. While intention to transport charcoal can and may be disclosed to municipalities when registering motorcycle ownership, we found that it is not typical to report intended usage of personal motorcycles. Charcoal burners may own or rent their motorcycles or hire a third party to handle this part of their value chain. Since charcoal is traded like any other commodity on the market, moving it from one place to another occasionally or 'on a request basis' means that one may not have to obtain permits for its transportation, specifically from the trader to consumers' premises. Motorcycles play a key role in transporting charcoal from the production sites to the main roads as well as from market places to final consumers who make bulk purchases.

Motor boats transport charcoal produced on Lake Victoria's islands, such as Kome, to the shore, from where it is delivered to busy market places, such as Gaba and Luzira (Figures 2 and 4e). The boats are manually loaded and unloaded by traders. Boat operators aim to arrive at the market early in the morning when trading centers are busy in anticipation of peak demand for charcoal. This means that at least part of the journey is made before dawn, which may affect taxing and transport permits for charcoal.

Trucks and lorries are mostly used for large quantities and long distances (Figure 4f). Even though forested areas are commonly perceived to be impassable by large vehicles, high-powered trucks and lorries can easily maneuver around young or less dense vegetation, making their way to remote charcoal production sites. In the Nwoya District, for example, we observed that charcoal burners cleared young vegetation near a concentration of earth-mound kilns to create way for large commercial vehicles and make loading convenient around the semi-forested area near the Naminato bridge. Therefore, in addition to the felling of trees, destruction of premature vegetation for the purpose of transportation should also be accounted for to fully assess the trail of destruction in obtaining this source of energy. Trucks set off for the city every evening, taking overnight trips to deliver the charcoal to markets by dusk. Indeed, Alem et al. [24], in their study in Ethiopia, also noted that charcoal deliveries to urban centers peak in the morning. We observed at various locations that those trucks en-route too early typically wait by roadsides in nearby towns (Figure 4f) before pursuing the remaining journey. Being lightweight, charcoal is over-loaded beyond legal and safe limits, for both economic and strategic

reasons, because fewer trips taken with higher loads per trip reduce frequency of trips, therefore eliminating costs of extra trips.

### 3.7. Charcoal as a Commodity

Beyond its purpose as an energy source, charcoal helps families out of abject poverty as a publicly traded commodity in local and export markets [6,15,21,33,45–48]. The charcoal supply chain has multitudes of stakeholders [23,43], including entrepreneurs, predominantly involved in skillful charcoal production; and middlemen specialized in trade, transport, and logistics [15]. In Uganda, we observed that buyers recognize the product easily without labels and sophisticated packaging, and therefore modern advertising is not common in this country. The quality of charcoal depends on tree species, as was also reported by Tabuti et al. [20] for firewood. Furthermore, we found that traders and consumers considered more subjective variables, such as density, as determinants of quality in charcoal. As various tree species are put through a single pyrolysis cycle and inventory is generally mixed up during packaging, identifying good quality charcoal is difficult at the conventional marketplace.

Most charcoal produced in Africa is consumed by the local population (see Shively et al. [43]). We observed cross-district trade, as documented by shipments via trucks or bicycles (see above and by the high variation in quality of charcoal in Kampala's market places, suggesting that it originated from various tree species common in different regions across the country. As charcoal moves from rural areas to populous urban centers to access bigger markets, varieties in its source and quality are more perceptible, especially to traders. Charcoal from the north is generally perceived to be of better quality because it is harder and burns slower since it is made from tree species, such as, *Albizia glaberrima* ("Nongo", Lowveld Albizia) sourced from Nwoya District, or *Combretum molle* ("Ndagi", Velvet Bushwillow) sourced from Kyankwanzi District (Northernmost part of the Central region). On the other hand, *Vernonia amygdalina* ("Mululuza", Bitter Leaf), *Spathodea campanulata* ("Kifabakazi", Tulip Tree/Nile Flame), *Artocarpus heterophyllus* ("Fene/Mufenensi", Jackfruit), and other species common in the Central region and the islands have softer stems, producing more lightweight and brittle charcoal, which depletes faster when ignited on open stoves.

Indeed, these variations set the basis for charcoal pricing practices. Market prices per sack are higher for charcoal of better quality, depending on its origin. We observed, for instance, that each 180–240 cm tall sack of fully processed charcoal from trees in the Northern districts may be priced at 65,000–85,000 Uganda Shillings (UGX), equivalent to US$17–23, whereas that from Central regions and the islands can fetch as low as 36,000–45,000 UGX (US$10–12). Beyond perceived quality, other variables that influence price include quantity purchased, negotiating power and skills, and location of final traders. For instance, we observed that charcoal burners in the Northern Nwoya District sold at 10,000 UGX (about US$3) per sack at the kiln, whereas the exact same charcoal fetches 65,000–85,000 UGX (US$17–23) per sack in urban centers.

However, in this largely informal trade, all unit and wholesale prices are negotiable, which means that it may not be uncommon to purchase 'inferior' quality at higher prices or the reverse. Even if standardized measurements are attempted to price non-bulk purchases, these vary by location, trader, or instruments used to measure. For instance, we found that some traders measure charcoal by "debe", a type of container [20], although these vary widely in size because they are mostly old recycled containers. Our observations further revealed that at no point along the supply chain is charcoal weighed on scales to determine its market value in Uganda. While producers, traders, and middlemen are more disposed to know origin, species and quality variations of charcoal, some end users have this knowledge too, depending on experience and the pursuit of good quality or at least 'value for money', especially where pricing differences are negligible because of variations in volumes purchased.

### 3.8. Charcoal and the Shared Economy

Because most parts of the charcoal supply chain are highly informal (see also Schure et al.) [29], collection of revenues at the national level is immensely complicated. Indeed, Sander, Gros, and

Peter [15] asserted that it contributes little to national accounts, costing for example the Tanzanian government over US$100 M annually in foregone taxes and licensing. These authors further attributed this revenue loss to the lack of fiscal and legal empowerment to enforce revenue collection in the central government. Khundi et al. [21] also pointed to the increasing challenges that decentralized district governments face in efforts to implement taxes and licenses to produce and transport charcoal in its upstream value chain.

Despite these challenges, however, our research showed that the trade of charcoal is not entirely tax-free. In decentralized market places, charcoal traders, like traders of other items, pay for occupancy at market stalls in the form of rents to administrators of local markets. Therefore, some traders have reserved spots in final decentralized market places, where they are acknowledged as operators in the downstream supply chain. Indeed, Van Beukering et al. [23] discussed that Tanzania's charcoal economy employed over 70,000 people and generated USD $200 million in 2002, and thus, it is not economically irrelevant. The economic importance of charcoal in developing countries has also been discussed by Mwampamba et al. [47].

Traders typically have integrated business processes with well-coordinated logistics to secure supply from rural areas [23,43]. This revenue is reinvested into services, such as maintenance and security of the local markets, which are major parts of small economies, with significant contributions to microeconomic development, or growth at the township level.

The key challenge here is that not all final traders sell at 'reserved spots'. As some operations of this supply chain are relationship-driven, knowing a rural supplier or urban consumers is sufficient to eliminate the need to operate within local infrastructure because trade can be done by phone, thereby diminishing rent as a business expense and increasing difficulty of tax collection.

We conclude that, although the economic 'pay-in' is substantially disproportionate along the value chain in contrast to other trades [15], tax or fee collection from charcoal is more likely to succeed in highly decentralized settings, where traders operate within designated market infrastructure, rather than on a national level.

Our research revealed that little to no standardization exists in the trade. For example, at kilns, waste and material inputs are not measured. However, output at this stage is recorded in number of sacks. All FAO [3] estimates for charcoal extraction and trade are in metric tons (Figure 1). In practice, however, charcoal in the sub-Saharan region is not sold by weight (see also Alem et al. [24]). At the points of sale, we observed that while wholesalers traded by the sack, smaller purchases are gauged for pricing in recycled metallic or plastic equipment [20], as also reported from other countries (e.g., Alem et al. [24]), whereby the types of containers used reflect materials available or trader's preferences. Traders use their judgment in pricing to determine what customers would pay for any given quantity. Final buyers take charcoal in disposable plastic bags of various sizes, leaving the traders' recycled equipment at market stalls for reuse as measuring tools. These common practices are out of circumstance, rather than predefined conditions or standards set by traders within the value chain.

*3.9. Handling and Storage for Charcoal as a Commodity*

The physical nature of charcoal is such that it is a hard but brittle and lightweight material, which is easy to store and handle [38]. Even though water, humidity or breakage do not destroy it, our investigations showed that charcoal is commonly preferred as dry pieces of small to medium sizes, because these are easy to set alight and fit well into charcoal stoves. Moreover, further breakdown into loose dust is unprofitable and poses a health risk, which influences how it is handled along the value chain. However, some traders turn charcoal dust into briquettes, which can be sold as well (Figure 5a). In some cases, we observed that the sacks with charcoal are covered to prevent wetting. While its lightweight nature invites overloading of trucks (Figure 4f), transporters ensure limited movement of the sacks during the journey in order to prevent disintegration.

### 3.10. Charcoal as an Aspect of the Household Energy Mix

Charcoal has always been the preferred fuel for cooking, and it has many attractive features, including its low to very low contents of ash, nitrogen, sulfur, and mercury, as well as its ease of storing and handling [38]. The IEA [10] has estimated that even today, 90% of households in 25 sub-Saharan countries rely on charcoal for cooking. Similarly, urban populations in many African countries use predominantly charcoal for cooking [29,49], with more than 80% of all urban households in sub-Saharan Africa relying on this fuel [33]. In Uganda, specifically, we found that charcoal consumption by many households is so highly ingrained in cooking processes that it persists even where modern alternatives such as biogas and electricity are present. Our observations are consistent with the results obtained by Drazu, Olweny, and Kazoora [19], who reported that 91% of grid connections in their sample were domestic, yet 89% of households still used charcoal. They further explained that there is a high level of autonomy in planning the energy mix of a household, which gives people the flexibility to plan for alternation between solid fuels and modern technologies, depending on a range of factors such as market prices, reliability, and convenience. For instance, in rural areas where more firewood is freely and readily available, charcoal is used less commonly [20], as also reported by Alfaro and Jones 2018 [26] for Liberia. In cities, people use electricity more selectively when its price increases, preferring to apply it mostly to light tasks, such as ironing and watching television rather than cooking [19].

We observed that, beyond its low price relative to that of electricity, charcoal is a common choice because it can be used more flexibly, where the consumers can gauge how much charcoal is needed to prepare meals and use quantities that are affordable to them. Even though Mukwaya [22] explains that some people that cannot afford charcoal would cook fewer meals per day, charcoal can be sold in very small quantities (e.g., as low as UGX 500 ($0.13) per 'debe', or container) [20], making it financially accessible to most people. Moreover, we found that firewood, which is a cheaper substitute, is widely sold alongside charcoal in marketplaces, such as, the Gaba Market at the Lake. Victoria shore or picked freely on private farms in rural areas [20]. Electric ovens have standard sizes that limit quantity of food prepared. When using charcoal, however, the quantity can be regulated according to family size especially because charcoal stoves (Figure 5b) come in different shapes and sizes, giving more flexibility to users in determining how much food is cooked at once depending on the population size of their households.

Our research revealed that compared to an electric oven or gas cooker, charcoal stoves are inexpensive. They are made from clay or recycled sheet metal (Figure 5b) and usually require no spare parts. Moreover, metallic stoves are typically painted by sellers to prevent rusting, thereby diminishing maintenance costs for the buyer. They are easy to lift and move outdoors, indoors, or to storage areas. These factors make it hard for modern energy alternatives to compete as fuel choices in households, despite being more convenient. Moreover, the highly erratic electricity supply from centralized energy distributors makes electric and gas ovens unreliable [19].

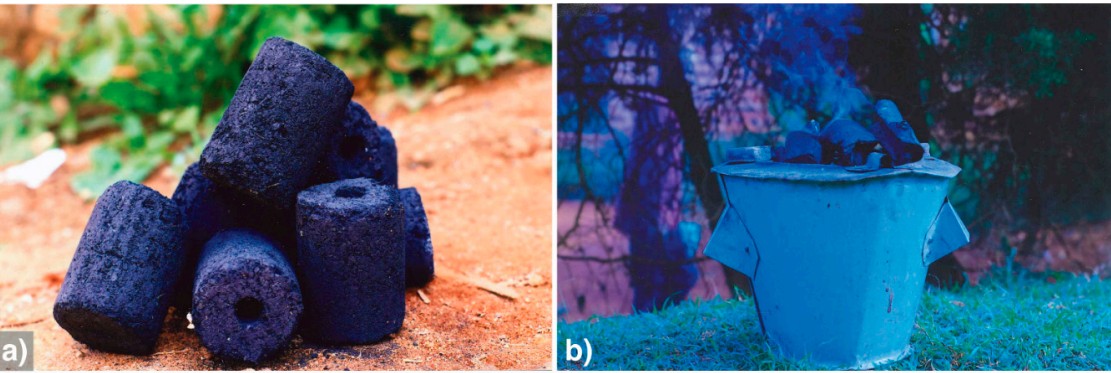

**Figure 5.** (**a**) Briquettes made of charcoal dust. Each briquette is about 5 cm long. (**b**) Charcoal cooking stove made of sheet metal in Nateete, a Kampala suburb.

### 3.11. Consumption and Health Implications

The current reliance on charcoal as main cooking fuel in sub-Saharan Africa [10,33,50] is expected to remain strong well into 2030 [51], because the fast increase in population and urbanization will lead to a further increase in the use of charcoal [33]. This trend must also be viewed from the perspective of the health impacts of both the production and the use of charcoal [45,52,53].

Charcoal stoves are simple, vary in size, and are made of different materials. They are portable and easy to handle (Figure 5b). We observed that, because of abundant air supply, charcoal stoves are often lit outdoors using matchsticks. However, during the rainy season and at night, the stoves are moved to verandas or indoor cooking areas to prevent rainwater from stopping the fire. We found that users do not commonly use protective equipment (e.g., gloves, overalls, masks), which can cause dirtying and burning from detached dust, and lead to inhalation of gases and particle emissions. As revealed by our field investigations, one must be constantly present to regulate the heat given off during cooking through adding charcoal pieces to the stoves for more heat or by adding layers of ash to manually reduce the heat or radiation given off. Once cooking is completed, one has to manually add ash or very small pieces of charcoal to stop the charcoal from burning or to preserve the heat for later use.

Charcoal is not child-safe where play areas overlap with cooking spaces, and its combustion elevates dangers of carbon monoxide poisoning [54]. The use of charcoal is further linked to premature death from inhalation of particulate matter [55,56], especially when used indoors. Moreover, as charcoal production does not represent complete combustion but rather a pyrolysis, high amounts of tarry vapors occur near the earth-mound kilns. These vapors contain a complex mix of organic compounds and non-condensable gases, including carbon monoxide ($CO$), carbon dioxide ($CO_2$), methane ($CH_4$), and heavier hydrocarbons [38]. The vapors also have serious impacts on the quality of the air and the water in the surrounding areas [38]. Bailis et al. [7] further report that the gases, particulate matter and polycyclic aromatic hydrocarbons (PAHs) given off around Brazil's charcoal kilns pose health hazards, especially to workers directly living at the sites.

Charcoal consumers generally know of these negative effects because some are noticeable during usage. For example, burning 'inferior'-quality charcoal typically releases substantial amounts of soot and smoke, which gives rise to visible deterioration of air quality, similar to when burning firewood (Figure 6), and leads to immediate irritation and short-term discomfort in the form of coughing and reddening of eyes. We conclude from our research that this is the reason why many consumers pursue 'good-quality' charcoal at the market place, although it is difficult to identify 'good' and 'bad' material. There are no guarantees for charcoal quality, however, due to high randomness of variables that influence its quality along the upstream value chain, such as sources, tree species, and skill of producers [11]. Moreover, "better quality" comes at higher prices, which may be unaffordable to people with lower disposable income [39].

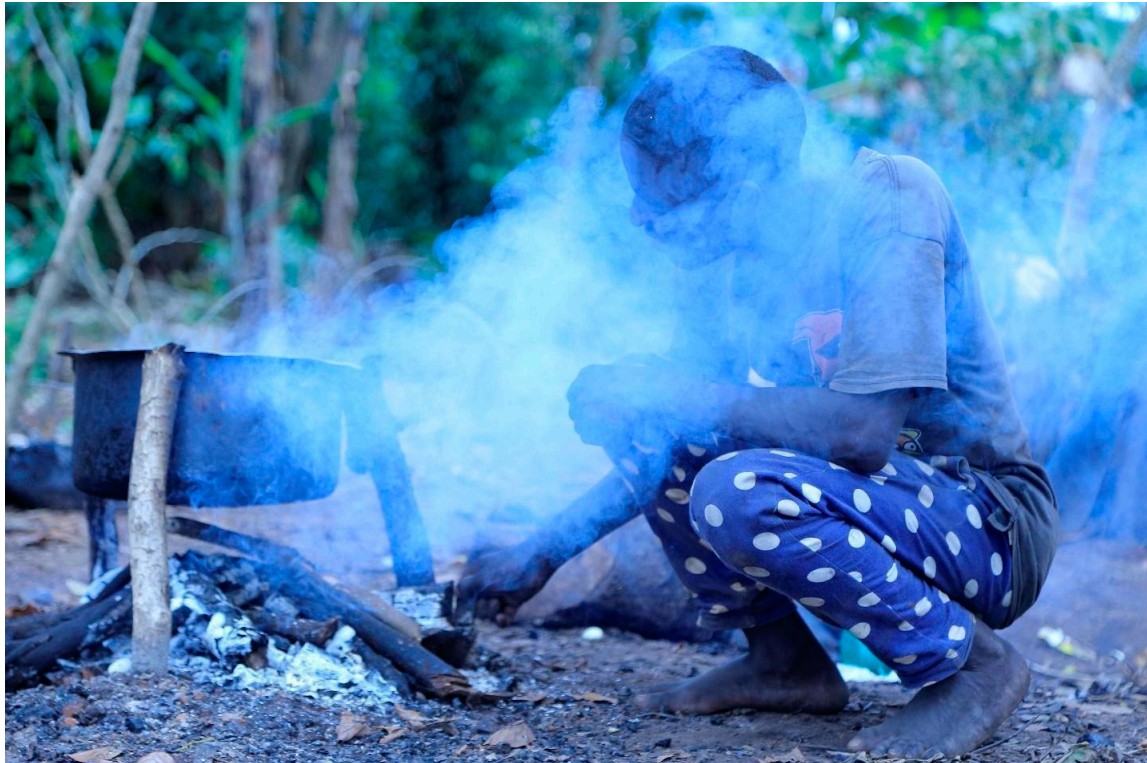

**Figure 6.** Cooking with biomass and deterioration of air quality. Near Kikumbi, in Mityana district.

### 3.12. Post-Consumption Residues

After complete combustion of charcoal, ash remains as a residue. It is a grey lightweight powdery substance with a propensity to float within the atmosphere. Even though we did find it at the earth-mound kilns, the main goal of the producers is pyrolysis, which does not generate large amounts of ash. Ash is therefore mainly accumulating at the consumer end, where the charcoal is burned to extract energy. Ohno and Erich [57] also showed wood ash may be a major source of nutrients, such as potassium, magnesium, calcium, and sodium. These chemical components of the ash depend on wood species burned, harvest season, and local soil types, and they can make ash a viable fertilizer without causing environmental concern [58].

## 4. General Comments on the International Charcoal Supply Chain

Operations of multitudes of stakeholders [15] are coordinated to bring charcoal to international markets at affordable prices. Although most charcoal is produced in Africa [3] and sold within the local markets, demand for charcoal has sustained worldwide and maintained the well-established import and export networks, which include traders that are not necessarily involved in upstream processes, such as raw material procurement and production [6]. The charcoal supply chain is characterized by informality [29], which leads to limited documentation [43], especially for the true origin and nature of the raw materials. Felling trees for the procurement of raw materials and production typically occur on rural private land [20,23,42], where charcoal burners and entrepreneurs agree on the use of trees and ownership of charcoal once the value-adding pyrolysis is completed. As charcoal changes hands from producers to initial bulk buyers or exporters, with limited sophistication of packaging materials—i.e., without labels, or barcodes for inventory management and tracking—information is lost about the source of logs, which makes it hard to detect its nature and quality for buyers. Additionally, opportunities are missed to provide warnings to consumers of the harmfulness in its use, such as the release of the lethal carbon monoxide, especially in informal markets where implementation of consumer regulation is futile [15].

Where charcoal is delivered directly to consumers who know producers, the main interest is usually in its price and functionality rather than sourcing methods. Charcoal is also sold further downstream to wholesalers who in turn sell it in smaller quantities to consumers in decentralized markets, such as, the Gaba market along the shore of Lake Victoria [4]. In these processes, the key interest of traders is to maintain reliable inventory levels, which secure supply to consumers and generate revenue. This is done in several ways, such as, through obtaining strategic information about a reliable upstream producer and supplier base, owning raw materials, recruiting charcoal burners for nomadic production, and finding landowners that permit charcoal production. However, these strategies depend on a range of factors, including knowledge of consumers and producers in the market, distances to production point [4], language proficiency, access to capital, regulatory conditions, and enforcement standards in areas where production occurs [15].

There is plenty of undocumented information among practitioners in the market place, and it is leveraged in many ways as a source of competitive advantage by traders. For instance, traders who know strategic details about their usually low-price suppliers are reluctant to share them to prevent arrests or being replaced as a middle man in the multi-tier upstream procurement. Still, some information is mentioned about the source of charcoal, especially where it bears economic advantages. For instance, a trader may state the imprecise origin of charcoal and species as a marketing strategy, since geographical origin of charcoal gives an impression of its quality to knowledgeable traders that may be procuring for more downstream trade or to consumers seeking 'value for money' for home use. Charcoal burners in rural areas, such as Mityana, Luwero, Nwoya, and Bulamogi [20], may be aware that final consumers in more urbanized areas pay higher prices for charcoal, but they lack the capital or resources to facilitate its logistics to areas with more favorable prices, which prompts them to sell their charcoal in bulk to enterprising traders who have the facilities to move products away from rural areas, which generally have lower demand, because of more affordable access to alternatives such as firewood, and low purchasing power [20]. The reluctance to share producer's information is in part due to stringent regulation such as regulatory bans that forbid the production process—e.g., in Nigeria and many Ugandan districts—which are enforced through arrests [6,20,23]. Traders may purchase wholesale or partial quantities and sell them within a profit margin to downstream buyers and consumers.

Despite the fact that production is illegal in some countries [6], the charcoal supply chain is sustained in part because transportation and consumption are legal worldwide, since charcoal is a legitimate product on local and international markets and a popular substitute to modern energy alternatives [39], for reasons beyond price alone. Indeed, its use is favorably linked to the taste of food [7,19,22,26]. Moreover, Alfaro and Jones [26] argued that charcoal consumption is so engrained in energy-use structures of some societies that it is unpragmatic to change the lifestyle. To a regulator, laws for the preservation of vegetation are crucial for the public interest of good health and a sustainable environment. In fact, this is the basis for bans and fines in the upstream supply chain [6]. The work of forest-focused environmental organizations is generally geared towards the preservation of public forests [15], even though more charcoal is produced in private rather than public forests [42], where government powerlessness is more perceptible [15]. Supply chains, whose products and 'survival' or raw material supply security are based on forest maintenance, are increasingly investing in forest management and leaning on certification entities to monitor forest loss and illegal logging, especially for markets where implications for company and product reputations are high [6]. However, raw materials are not typically sourced where such formalities are the norm. Indeed, in local markets where trade is legal and more common, sale of charcoal is not sensitive to a seller's 'environmental reputation', since charcoal (and energy in general) has a demand that is relatively stable if not higher despite disproportionate price increases.

This reality, as in many supply chains, is a complex web of events, with multi-tiered buyers, sellers, importers, and exporters, who may also purchase from each other depending on inventory cycles, market prices, location, distance to buyers, or consumer requirements. In market places,

consumer/trader priorities, such as the pursuit of low price and good quality, are reflected in their purchasing decisions. The common goal for each player is usually to maximize revenue or to procure functional charcoal at the minimum price. For charcoal producers close to the earth-mound kilns, profit can be secondary to survival. Some consumers in urban areas also view low prices as a key priority because it determines the number of meals per day [22], whereas others use charcoal mainly for leisure.

## 5. Summary and Conclusions

As a legally traded commodity in several markets, charcoal has sustained its significance worldwide, thus encouraging production even in places with scarce vegetation and where deforestation to procure its raw materials and its production are illegal. The persistence of charcoal demand, its global trade over the past 20 years, and the predicted increase in charcoal use document that modern alternatives have not successfully replaced its utility as an energy resource. While the informal charcoal economy has socioeconomic advantages, such as entrepreneurship and local government revenue from permits, which relieve its stakeholders from abject poverty and generate microeconomic development, the production and use of charcoal worldwide pose risks to the environment and to public health. Highly random and decentralized production, consumption, and trade practices thwart efforts towards effective centralized forest management and economic planning. Moreover, they make management of waste and air quality difficult for all stakeholders.

In practice, understanding the complexities for sourcing of raw materials in supply chains that span international boundaries is crucial for various reasons: (1) Because it helps to gauge the opportunities and limits of policy. As government power is not absolute, it would also help planners to determine the nature of programs likely to persuade landowners to preserve private forests despite barriers, such as, lack of interest and funding. (2) While charcoal consumption is global, the more evident fact is the disproportionate location of upstream procurement activities. In this supply chain, producers and traders have strategically directed their pursuit of raw materials to less regulated regions, or where strong regulation exists but with extreme difficulty in enforcement. (3) To a great extent, the supply of charcoal and the destructive environmental trail of its sourcing practices are still sustained by its global demand, yet in many markets, consumers are unaware of the damage of upstream and downstream activities involved in sourcing and consuming charcoal. Even though the growing demand for charcoal increases opportunities for both rural and urban populations and reduces poverty, the current and predicted levels of charcoal use in many developing countries, especially in sub-Saharan Africa, pose a threat to the long-term sustainability of natural forests. The increased use of charcoal further leads to environmental degradation and may cause severe health impacts, particularly when charcoal is burned indoors.

**Author Contributions:** C.N. and R.G. conceptualized and designed the research project. C.N. carried out all field work and interviews. Both authors contributed to the writing and revising of the manuscript.

**Funding:** This research project was supported in part by a travel grant obtained from the University of Pennsylvania.

**Acknowledgments:** We would like to thank the charcoal producers and landowners that allowed us access to their premises and agreed to the interviews. In addition, we are grateful to Yvette Bordeaux, Edward Kiwanuka, Rebecca Nansamba, Rebecca Eseza Nabitoogo, Edgma Calvin Luutu, and Fred Wamala for various support during this study. We thank the three reviewers and the editors for their valuable comments and suggestions, which helped us in improving the manuscript.

**Conflicts of Interest:** The authors declare no conflict of interest. The funders had no role in the design of the study; in the collection and interpretation of the field data; in the writing of the manuscript, or in the decision to publish the results of our investigation.

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
