# Peer review of "Charcoal as an Energy Resource: Global Trade, Production and Socioeconomic Practices Observed in Uganda"

_resources, doi:10.3390/resources8040183_

Round 1

Reviewer 1 Report

The format of the Reference of this manuscript does not conform to that of Resources. The authors should modify them according to the instructions of the journal. Moreover, there are not any citation of Resources itself.

The Introduction of the paper is mainly based on the general introduction of the background, lacking the combing of the relevant literature on the subject of the study. It is recommended that the author rewrite this section in academic style and include the corresponding references [1-3].

The content and format of this paper are poorly arranged - such as the lack of Conclusions, and the format of the 3.12 and previous titles are not the same, etc. It is highly recommended that the authors carefully arrange the content of the paper according to the template of Resources.

[1] Efficiency evaluation of industrial waste gas control in China: A study based on data envelopment analysis (DEA) model. Journal of Cleaner Production 2018179, 1–11. 

[2] Evaluating China’s Air Pollution Control Policy with Extended AQI Indicator System: Example of the Beijing-Tianjin-Hebei Region. Sustainability 201911(3), 939.

[3] Analysis of the Air Quality and the Effect of Governance Policies in China’s Pearl River Delta, 2015–2018. Atmosphere 201910(7), 412.

Author Response

The format of the Reference of this manuscript does not conform to that of Resources. The authors should modify them according to the instructions of the journal. Moreover, there are not any citation of Resources itself.

Modified: All full names of organizations are listed, e.g., from FAO, IEA, MEMD, UBOS. Otherwise, we believe that the formatting corresponds to the “Resources” format (see email from Madeline Zhang, MDPI from October 7, 2019)

The Introduction of the paper is mainly based on the general introduction of the background, lacking the combing of the relevant literature on the subject of the study. It is recommended that the author rewrite this section in academic style and include the corresponding references [1-3].

We do not quite understand this comment, because that is exactly what we believe our introduction is doing…. We presented an overview of the current state of knowledge and provided a compilation with the relevant literature, all tied together in an introduction.  The second comment, i.e., to include three specific references suggested by the reviewer represents a concern to us, for two reasons:

None of the three suggested articles has anything to do with the topic of our manuscript. Moreover, our careful reading of the suggested papers revealed that the word “charcoal” is not even mentioned once in any of the papers. We do not understand why they were suggested. All suggested references are by the same author or group of authors, which indicates that the reviewer wants us to cite his/her papers. This is, in our opinion, not appropriate, even more so when the papers have absolutely no connection to the main of the research topic we would like to publish.

The content and format of this paper are poorly arranged - such as the lack of Conclusions, and the format of the 3.12 and previous titles are not the same, etc. It is highly recommended that the authors carefully arrange the content of the paper according to the template of Resources.

In the previous copy, 3.12 was in bold (non-Italic) letters (all other sections titles were not). It seemed to us like a standalone “section” (but not a conclusions section). Formatting is now uniform.  We do like the idea of a “Conclusions” section, but refrained from doing so because it was not suggested in the formatting instructions provided by MDPI.  We now have restructured the manuscript and included a Conclusions Section.

[1] Efficiency evaluation of industrial waste gas control in China: A study based on data envelopment analysis (DEA) model. Journal of Cleaner Production 2018179, 1–11.  No mention of “charcoal” in this paper and no thematic connection whatsoever to our manuscript.

[2] Evaluating China’s Air Pollution Control Policy with Extended AQI Indicator System: Example of the Beijing-Tianjin-Hebei Region. Sustainability 201911(3), 939. No mention of “charcoal” in this paper and no thematic connection whatsoever to our manuscript.

[3] Analysis of the Air Quality and the Effect of Governance Policies in China’s Pearl River Delta, 2015–2018. Atmosphere 201910(7), 412. No mention of “charcoal” in this paper and no thematic connection whatsoever to our manuscript.

Reviewer 2 Report

The manuscript is interesting and well written but I have some comments that must be gone through.

First of all, materials and methods are not clear enough. It is written that many people were interviewed to collect data, but much more details should be provided: how many people were interviewed? what was their role in the production chain? etc.

in lines 126-129 it is written that the different sites had several differences in terms of vegetation, what are these differences? A detailed description of the context should be provided.

I suggest moving paragraphs 3.2 and 3.3 to some other section, such as in materials and methods, because they are not results of the study. 

The results section should be revisited because are reported results without explanation of the calculation method. See lines 414-422. This is a market analysis with no background and no reference to data collection. It is not a result. 

lines 374-378 should be rewritten.

lines 456-459 are about conclusions so should be moved.

An effective section for conclusions is missing and should be properly introduced.

In general, the manuscript is much descriptive and could be shortened.

Author Response

The manuscript is interesting and well written, but I have some comments that must be gone through. First of all, materials and methods are not clear enough. It is written that many people were interviewed to collect data, but much more details should be provided: how many people were interviewed? what was their role in the production chain? etc.

Thank you for this very valuable comment, which we highly appreciate. We have followed the recommendation and now provide a Table, which we included as a Supplementary Material (Table S1). The entire section “Supplementary Materials” is new and was added as a result of this comment.

in lines 126-129 it is written that the different sites had several differences in terms of vegetation, what are these differences? A detailed description of the context should be provided.

Again, an excellent suggestion. Thank you for this very valuable comment, which we highly appreciate. We have followed the recommendation and now provide a Table, which we included as a Supplementary Material (Table S2). The entire section “Supplementary Materials” is new and was added as a result of this and the previous comment.

I suggest moving paragraphs 3.2 and 3.3 to some other section, such as in materials and methods, because they are not results of the study. 

Thank you for these suggestions. However, we do not agree with the reviewer’s assessments, as explained below:

Paragraphs 3.2 and 3.3 are part of the results. Paragraph 3.2 discusses what we observed during our field work in Nwoya’s private forests. Indeed, Naminato bridge in Nwoya is located near Murchison Falls National Park, which has wildlife (monkeys are untethered because they are not dangerous). The charcoal burners at the study site constructed this bridge with basic materials to cross a small tributary that flows from Murchison, so, the pests cross freely to farmlands.

Our interviewees from Kome Island were met in Gaba at the shore of Lake Victoria (see our new, added Table S1) and the results discussed are directly derived from their experience. Ref [30] is added to show an increasing trend in Rice farming that was reported in the local newspaper.

Paragraph 3.3 is a description of processes used by the charcoal burners as observed during our study, and thus, is a result. Naming of these processes is consistent with their local language and details are based on their experiences (Photos in Figure 3)

The results section should be revisited because are reported results without explanation of the calculation method. See lines 414-422. This is a market analysis with no background and no reference to data collection. It is not a result. 

Lines 414-422 are not conclusions derived from our calculation. Rather, they are price ranges given by interviewees (see our new, added Table S1) that have actively participated in the trade or downstream part of the supply chain on a regular basis. Charcoal burners in the upstream also participated in whole-sale trade (Fig. 4 c and b) from near the kiln, and the prices given are specific findings of our study, with interviewees’ judgement based on typical prevailing conditions of the informal markets by locations. Figure 4 (b, c, d and f) feature typical examples of sacks we observed on the market (line 414)

lines 374-378 should be rewritten.

Thank you for pointing this out. We have changed it.

lines 456-459 are about conclusions so should be moved.

Thank you for this excellent suggestion. As a result, we now have included a “Conclusions” section.

An effective section for conclusions is missing and should be properly introduced.

Thank you for this excellent suggestion. As a result, we now have included a “Conclusions” section.

In general, the manuscript is much descriptive and could be shortened.

We do not agree with this recommendation, because we made it as short as possible without omitting important details. Moreover, there is no word limit.

Reviewer 3 Report

The authors presented a very interesting article. I really enjoyed a lot to read it. This is my major research area, and for that reason I really enjoy to see people presenting their experiences concerning the charcoal production.

I only can suggest the authors to include a few more references, mainly from European countries where a lot of research is occurring about this subject.

It would be a very improvement to refer that charcoal production, if using sustainable raw materials without forest depletion, can be considered a NET (negative emissions technologies), when the resulting material is incorporated to soils.

In my opinion will be ready to be published after the revision of these minor questions.

Author Response

The authors presented a very interesting article. I really enjoyed a lot to read it. This is my major research area, and for that reason I really enjoy to see people presenting their experiences concerning the charcoal production.

Thank you for your comment, which we appreciate even more, because the reviewer is a specialist in this area of research. His/her opinion and positive evaluation, therefore, are especially meaningful to us.

I only can suggest the authors to include a few more references, mainly from European countries where a lot of research is occurring about this subject.

Thank you for this suggestion. We assume the reviewer refers to biochar, which is produced to extract energy as well as to produce material that can be used as soil amendment. While this is an important aspect of charcoal, or specifically biochar, it is not relevant at all in the study area, where charcoal is produced exclusively to serve as a fuel. However, to accommodate the reviewer comment and address this this issue, we added a sentence and a reference to guide the reader to the topic of biochar (new Line 321). 

It would be a very improvement to refer that charcoal production, if using sustainable raw materials without forest depletion, can be considered a NET (negative emissions technologies), when the resulting material is incorporated to soils.

This is interesting, but again refers to biochar, which is irrelevant for this study in Uganda, where charcoal is produced almost exclusively to be used as a fuel (see above). 

In my opinion will be ready to be published after the revision of these minor questions.

We are grateful to this reviewer for the very positive evaluation and for the excellent suggestions

Round 2

Reviewer 1 Report

Unfortunately, the authors did not make changes in accordance with the review comments. More important to be noted is that in the 42 references cited by the authors, 26 were from web files, not academic papers, and none of them came from Resources. In view of this, I think the article is not suitable for publication in Resources.

Author Response

Unfortunately, the authors did not make changes in accordance with the review comments. More important to be noted is that in the 42 references cited by the authors, 26 were from web files, not academic papers, and none of them came from Resources. In view of this, I think the article is not suitable for publication in Resources.

In the first round, Reviewer 1 requested that we include his/her own papers, which however, have absolutely no relevance to our manuscript: As stated in our previous response, the papers suggested by this reviewer (from round 1) do not contain any information on the subject and further, the word “charcoal” does not even occur in those papers. Therefore, citing such papers is not appropriate, and we are very surprised that the reviewer wants us to include his/her own papers knowing that they have no relevance.

Now, in the second round of reviews, Reviewer 1 suggests that we use too many on-line resources as references, again a totally inappropriate comment. As expected for such a topic, it is necessary to use a diverse range of sources, including reports from the United Nations Food and Agricultural Organization (FAO), World Bank, International Energy Agency, and official government websites (e.g., Rural Electrification Agency and the Bureau of Statistics from the Ugandan government).

These resources are official publications, produced by highly respected government or international entities, which of course do not publish in scientific journals, such as, Resources. In addition, it is inappropriate to demand that we cite papers from Resources. This is – correctly, according to international publishing standards – Not mandated by the Resources’ instructions for authors. By-the-way, we do cite papers from some of the other MDPI journals (e.g., Sustainability). As our article discusses a diverse range of resource subjects, i.e., natural, energy, land, and ecology resources, which are listed on the MDPI website for the journal’s Aims and Scope, we believe that it is highly suitable for publication in resources.

Reviewer 2 Report

The authors did a quite good job in giving information about the materials and methods in the supplementary material, but no change was made in the text.

I still believe the manuscript should be improved before publication because all interesting data about how the study was performed (materials and methods) are written in this supplementary material. Therefore, I suggest that authors keep this section but also make a brief summary of these 3 tables and introduce the most important data into the text, as table or figure (e.g., number of people interviewed per district and their role, the main general characteristics of the vegetation, as well as a brief discussion on the alternative uses of vegetation for local people). 

Finally, I still find that the section of results is too much descriptive and I recommend introducing sentences such as "from this research", "as results from the interviews" etc. in order to make the reader aware that what is written derives from the analysis of the collected data, otherwise it is not very clear.

Author Response

I still believe the manuscript should be improved before publication because all interesting data about how the study was performed (materials and methods) are written in this supplementary material. Therefore, I suggest that authors keep this section but also make a brief summary of these 3 tables and introduce the most important data into the text, as table or figure (e.g., number of people interviewed per district and their role, the main general characteristics of the vegetation, as well as a brief discussion on the alternative uses of vegetation for local people). 

We would like to remind both the editor and the reviewer that, after the first round of reviews, we added a large amount of information as “Supplementary Materials” (Tables S1, S2, S3”). This was done in direct response to the reviewer’s excellent suggestion and because we believe that the manuscript in its first version lacked that crucial information. We expressed our gratitude to the reviewer 2 of the first round of reviews. As this supplementary information is more than 10 pages long, listing all the requested details, it is too long to be included within the main text. While this information is important, providing all the background information for our study, we are convinced that adding all 3 Tables to the main text would take away from what we show to the reader as the ‘scope’ of the article, i.e., Global Trade, Production and Socioeconomic Practices. We therefore decided to add a summary table to the main text (new Table 1) in order to accommodate the reviewer’s additional request. With that, we believe we satisfied this reviewer comment.

Finally, I still find that the section of results is too much descriptive and I recommend introducing sentences such as "from this research", "as results from the interviews" etc. in order to make the reader aware that what is written derives from the analysis of the collected data, otherwise it is not very clear.

Thank you very much for this observation. We have included the words reviewer 2 has recommended to the text and believe that this distinction is now clear to the reader as a result.